# Minocycline Attenuates Sevoflurane-Induced Postoperative Cognitive Dysfunction in Aged Mice by Suppressing Hippocampal Apoptosis and the Notch Signaling Pathway-Mediated Neuroinflammation

**DOI:** 10.3390/brainsci13030512

**Published:** 2023-03-19

**Authors:** Junjie Liang, Shanshan Han, Chao Ye, Haimeng Zhu, Jiajun Wu, Yunjuan Nie, Gaoshang Chai, Peng Zhao, Dengxin Zhang

**Affiliations:** 1Department of Anesthesiology, Wuxi Maternal and Child Health Care Hospital Affiliated to Jiangnan University, Wuxi 214002, China; 2Department of Basic Medicine, Wuxi School of Medicine, Jiangnan University, Wuxi 214122, China

**Keywords:** POCD, sevoflurane, minocycline, apoptosis, neuroinflammation, Notch signaling pathway

## Abstract

Postoperative cognitive dysfunction (POCD), an important postoperative neurological complication, is very common and has an elevated incidence in elderly patients. Sevoflurane, an inhaled anesthetic, has been demonstrated to be associated with POCD in both clinical and animal studies. However, how to prevent POCD remains unclear. Minocycline, a commonly used antibiotic can cross the blood-brain barrier and exert an inhibitory effect on inflammation in the central nervous system. The present work aimed to examine the protective effect and mechanism of minocycline on sevoflurane-induced POCD in aged mice. We found that 3% sevoflurane administered 2 h a day for 3 consecutive days led to cognitive impairment in aged animals. Further investigation revealed that sevoflurane impaired synapse plasticity by causing apoptosis and neuroinflammation and thus induced cognitive dysfunction. However, minocycline pretreatment (50 mg/kg, i.p, 1 h prior to sevoflurane exposure) significantly attenuated learning and memory impairments associated with sevoflurane in aged animals by suppressing apoptosis and neuroinflammation. Moreover, a mechanistic analysis showed that minocycline suppressed sevoflurane-triggered neuroinflammation by inhibiting Notch signaling. Similar results were also obtained in vitro. Collectively, these findings suggested minocycline may be an effective drug for the prevention of sevoflurane-induced POCD in elderly patients.

## 1. Introduction

As society ages, more than 50% of all elderly individuals are estimated to undergo at least one surgical procedure [1]. Thus, a growing number of studies are focusing on the effects of surgery and/or anesthesia on brain function in aged patients. Postoperative delirium (POD) is an acute neurological complication characterized by decreased environmental awareness and attention deficit that occurs early after surgery and affects nearly 60% of elderly individuals [2]. Postoperative cognitive dysfunction (POCD), featuring memory deficit, reduced capability of processing information, and anxiety, is a further exacerbation of POD [3]. Sevoflurane, a frequently utilized inhalation anesthetic in clinics, has been shown to be strongly associated with POCD. Chai et al. suggested that sevoflurane enhances the expression of acidic leucine-rich nuclear phosphoprotein-32A (ANP32A, the key component of inhibitors of acetyltransferases) by inducing transcription factor CCAAT/enhancer binding protein beta (C/EBPβ), which may inhibit histone acetylation and promote cognitive impairment in aged mice [4]. Yu and colleagues demonstrated that repeated sevoflurane administration increases the concentrations of NUAK family SNF1-like kinase 1 (Nuak1, an AMPK-related kinase) in neonatal mice, which could induce brain Tau phosphorylation and lead to cognitive dysfunction [5]. Wang and collaborators found that sevoflurane inhalation might induce cognitive deficit in young mice via autophagy induction in the hippocampus [6]. Many studies have examined sevoflurane-induced POCD, demonstrating that sevoflurane causes POCD through a variety of mechanisms in mice of different ages. However, the exact underpinning mechanism remains undefined, making it a hot topic for many anesthesiologists.

Inflammation is the initial stage in multiple pathological processes. Inflammation in the hippocampus, the major brain area related to learning and memory, negatively affects cognitive ability. Sevoflurane has been shown to cause central nervous system inflammation. Qin et al. found that sevoflurane reduces immune response in young rats [7]. Dong et al. demonstrated that sevoflurane enhances the generation of the proinflammatory cytokine IL-6 as well as cognitive dysfunction by inducing Tau migration from neurons to microglia [8]. Li et al. suggested that sevoflurane induces NLRP3 inflammasome formation and reduces brain-derived neurotrophic factor (BDNF) production to promote neurodegeneration and neuroinflammation [9]. Gui et al. found that long-term inhalation of sevoflurane and surgery could induce neuroinflammation, down-regulate glial cell-derived neurotrophic factor (GDNF), and reduce the level of neurogenesis in the hippocampus, which contributed to cognitive dysfunction in neonatal rats [10]. Notch is a conserved pathway controlling cell fate. Previously reported findings indicated that Notch signaling is tightly associated with tumor development [11]. Recently, it was demonstrated that aberrant Notch pathway induction is strongly associated with inflammatory events, e.g., allergic airway inflammation [12] and renal inflammation [13]. However, whether the Notch pathway participates in hippocampal inflammation induced by sevoflurane remains unknown.

Minocycline, a second-generation tetracycline belonging to broad-spectrum antibacterial drugs, has high effective tissue permeability and antibacterial activity. Minocycline has been reported to exert a variety of pharmacological effects that are different from its antimicrobial activity, e.g., anti-inflammatory and antiapoptotic activities, in the central nervous system [14,15]. Tian et al. found that minocycline plays a protective role in hippocampal neurons of aged rats by alleviating sevoflurane-induced apoptotic death, inflammatory response, and beta amyloid (Aβ) production. [16]. Recently, Yang et al. found minocycline ameliorates diabetic neuropathic pain (DNP) by downregulating Notch and inactivating Notch signaling in microglia in the spinal cord [17]. However, whether minocycline attenuates sevoflurane-induced neuroinflammation by modulating Notch signaling remains unknown.

Based on the above findings, we hypothesized that the Notch signaling pathway plays a critical role in sevoflurane-induced neuroinflammation, and minocycline may alleviate such inflammation by inhibiting excessive activation of Notch signaling, thereby improving cognitive impairment. Consequently, the present work aimed to examine the role of Notch signaling in sevoflurane-associated neuroinflammation and to determine the therapeutic impact of minocycline.

## 2. Materials and Methods

### 2.1. Animals

Sixty-three specific pathogen-free (SPF) male C57BL/6J mice aged eighteen months (Changzhou Cavens Experimental Animal Co., Ltd., Changzhou, China) were utilized throughout this study. Mouse housing was carried out in a specific pathogen-free environment with a 12-h photoperiod and full access to food and water. The experimental protocol had approval from the Animal Ethics Committee of Jiangnan University (JN. No. 20201230c1401231[379]). We used as few animals as possible for experiments and minimized the severity of their suffering. After one week of adaptation, mice were randomized into the control (Con), sevoflurane (Sev), and sevoflurane+minocycline (Sev + Min) groups, with 21 animals/group. In each group, 10 mice were used for the Morris water maze and fear conditioning test, 8 were used for immunofluorescence, and 3 were used for LTP. After the behavior tests, anesthesia was carried out with 2% sodium pentobarbital solution at 80 mg/kg and the animals were sacrificed. The hippocampal tissue was isolated at 4 °C and kept at −80 °C for subsequent assays.

### 2.2. Minocycline Treatment and Sevoflurane Exposure

The sevoflurane anesthesia procedure was carried out as previously reported [4]. In brief, mice were placed into a transparent airtight box and administered 95% oxygen and 3% sevoflurane (Shanghai Hengrui Pharmaceutical, Shanghai, China) by inhalation at a flow rate of 1–2 L/min on an anesthesia machine (R620-S1, RWD Life Technology, Shenzhen, China) for two hours daily for three consecutive days, while control mice received 95% oxygen under the same environmental conditions. Animal body temperature was kept at 38 °C throughout the entire anesthesia procedure using a heating blanket. After the end of the sevoflurane anesthesia, mice were transferred to a new cage maintained at room temperature and ambient air until free movements occurred. Hydrochloride minocycline was obtained from MCE (MedChemExpress, Monmouth Junction, NJ, USA) and dissolved in saline. One hour before sevoflurane exposure, minocycline was administered by intraperitoneal injection in mice at 50 mg/kg according to a previous study [18], the animal in the Con and Sev groups were intraperitoneally administered equivalent amounts of saline.

### 2.3. Cell Culture and Sevoflurane Treatment

BV2 microglia and HT-22 cells were utilized as representative microglial cells and hippocampal neurons, respectively. Mouse hippocampal HT-22 cells, as a cell model, were obtained from mouse hippocampus and have been widely used in many neural system disease studies. BV2 microglia cells were obtained from immortalized mouse microglial cell lines showing inflammation and phagocytic feature upon induction [19]. Both HT-22 and BV2 cells were provided by Procell Life Science & Technology (Wuhan, China). Cell culture was performed at 37 °C in a humid environment containing 5% CO_2_ in DMEM (Cytiva, Shanghai, China) with antibiotics (penicillin [100 units/mL] and streptomycin [100 g/mL], Beyotime, Shanghai, China) and 10% FBS (BIOEXPLORER, Carolina, USA). The sevoflurane treatment protocol was described previously [20]. After 48 h, cells in culture dishes were randomized into the control, sevoflurane, minocycline (50 μM), and sevoflurane plus minocycline (1, 10, and 50 μM) groups. Culture dishes underwent incubation in an anesthesia induction chamber at 37 °C (RWD Life Science, Shenzhen, China) with a freshly prepared gas mixture (21% O_2_, 5% CO_2,_ and 69% N_2_) or 3% sevoflurane for 6 h. Minocycline was added to the medium 1 h before sevoflurane exposure.

### 2.4. Bromodeoxyuridine (BrdU)-Labeling

BrdU was obtained from Sigma-Aldrich Trading (China) and intraperitoneally injected into mice at the beginning and end of sevoflurane inhalation in each group while establishing the sevoflurane anesthesia model. The BrdU powder was dissolved in 37 °C saline, and the injection dose was 100 mg/kg b.i.d for three days.

### 2.5. Morris Water Maze (MWM) Test

Thirty mice (*n* = 10) were assessed for learning and memory function after the final day of sevoflurane exposure. The MWM experimental method was described previously [21]. Briefly, the MWM system consisted of a circular tank filled with water that appeared white with milk powder, and the water temperature was kept at 19–20 °C. The tank was evenly split into four quadrants, of which one contained the platform placed 1 cm below the water surface. After receiving sevoflurane anesthesia, mice underwent training with four trials daily for five days. Two consecutive trials were separated by a minimum time interval of 15 min. To perform a trial, the animals were placed in four distinct quadrants of the water maze sequentially and allowed 60 s to freely find the hidden platform; otherwise, they were directed toward the platform and forced to remain on it for 30 s. On day 6, mice underwent the spatial probe test, where they were tested without the platform present. Escape latency, the target time ratio, the swimming track, and the platform crossing times were recorded using a WMT-100 Morris water maze video tracking system.

### 2.6. Fear Conditioning Assay

The fear conditioning assay procedure was described previously [22]. Briefly, after sevoflurane or air exposure, the animals were positioned in a box containing a stainless-steel grid floor for foot shock and allowed to adapt to the environment for 3 min. In cued fear conditioning (day 1 after sevoflurane or air exposure), the mice were submitted to a sound stimulation at 70 dB for 30 s, followed by an electrical stimulation at 1 mA for 2 s, a cycle that was repeated 3 times. During this period, the percentage of freezing time in mice was determined. In the cued fear memory test (5 days after cued fear conditioning), the animals were positioned in a box for 3 min, followed by exposure to only 70 dB sound stimulation for 30 s three times. Memory in the cued fear assay was determined as the percentage of time that the animals remained frozen over the course of the three trials (Figure 1F).

### 2.7. Brain Immunofluorescence

Brains underwent fixation with 4% paraformaldehyde overnight, followed by transfer to 20% and 30%sucrose in PBS for 2 days and 3 days, respectively. After embedding in the Tissue-Tek OCT compound (Sakura, Tokyo, Japan), brain sections underwent 25-µm sectioning using a vibratome slicer (Leica, Nussloch, Germany). These sections underwent permeabilization with 0.5% Triton X-100 dissolved in PBS for 30 min at ambient temperature to dissolve the phospholipid membrane and blocking with 5% bovine serum albumin (BSA) for 1 h to block nonspecific sites. Incubation with primary antibodies was performed at 4 °C overnight. Following three PBS washes, subsequent incubation was carried out with secondary antibodies for 1 h at 37 °C. A microscope (Carl Zeiss, Freistaat Thüringen, Germany) was utilized for imaging. ImageJ (version 1.53m) was employed to assess positive cells in the hippocampal DG region of a given section, and then the numbers of positive cells in all sections from the same mouse were added for statistical analysis. Table 1 lists all primary antibodies applied in immunofluorescence.

### 2.8. Cell Immunofluorescence

Cells (5 × 10^4^ cells/well) seeded in 24-well dished with cover slips underwent overnight incubation and exposure to minocycline (1, 10, and 50 μM) 1 h before incubation with 3% sevoflurane for 6 h. Next, a 10-min fixation was carried out with 4% paraformaldehyde, and cells were permeabilized with 0.5% Triton X-100 in PBS. Blocking (5% BSA, 1 h) was followed by successive incubations with primary (overnight, 4 °C) and secondary (2 h, ambient). Finally, DAPI (2.5 μg/mL) counterstaining was carried out before analysis under a fluorescence microscope (Carl Zeiss). Table 1 lists all primary antibodies applied for immunofluorescence.

### 2.9. TUNEL-Based Cell Apoptosis Analysis

Cells (1 × 10^4^/well) seeded in 96-well plates with cover slips underwent overnight incubation and treatment with minocycline (1, 10, and 50 μM) for 1 h before induction with 3% sevoflurane for 6 h. According to the TUNEL FITC detection apoptosis kit (Vazyme, Nanjing, China) protocol, cells underwent fixation with 4% formaldehyde (25 min) and permeabilization with 0.2% Triton X-100 PBS (5 min). Next, 1× equilibration buffer was equilibrated for 10–30 min at ambient. This was followed by a 1 h with TdT incubation buffer at 37 °C. DAPI (2.5 μg/ML) counterstaining was carried out, followed by fluorescence microscopy. Cells can be divided into two groups using this technique: living (extremely low background fluorescence) and apoptotic (strong green, fluorescent signals) cells.

### 2.10. Western Blotting

Hippocampal tissue specimens or cells underwent lysis with RIPA buffer (Beyotime, Shanghai, China) plus 1 mmol/L PMSF. Proteins separation utilized 12% SDS-PAGE gels (Vazyme, Nanjing, China) for approximately 1 h, followed by transfer onto nitrocellulose membranes. After a 1 h blocking (with 5% BSA in Tris-buffered saline Tween-20 (TBST), the samples were submitted to successive incubation) with primary (overnight, 4 °C) and secondary (1 h, ambient) antibodies. Finally, these bands were detected on a Tanon-2500B chemiluminescence imager (Shanghai Tanon Technology, Shanghai, China), and ImageJ was utilized for densiometric quantitation. The target protein’s expression was standardized to total β-actin levels. Table 2 lists all primary antibodies utilized for Western blotting.

### 2.11. Reverse Transcription and Real-Time Quantitative PCR

Total RNA extraction was carried out from BV2 microglia cells using with TRIzol reagent. Reverse transcription employed Hifair^®^ III 1st Strand cDNA Synthesis SuperMix for qPCR (Yeasen, Shanghai, China) with primer sets for iNOS, IL-1β, and GAPDH. 2^−△△Ct^ method was utilized for the analysis of data after GAPDH normalization. Table 3 lists the primers utilized for PCR.

### 2.12. Electrophysiological Analysis

Mice were anesthetized by intraperitoneally injecting 2% sodium pentobarbital at 80 mg/kg and then sacrificed. The brain was obtained after decapitation and placed in artificial cerebrospinal fluid (ACSF) saturated with mixed gases (95% O_2_ and 5% CO_2_) at 0~4 °C. After cooling the brain tissue to 0~4 °C, we dissected the hippocampus, which was sliced along the sagittal plane with a 400 µm thick vibrating slicer. Then, hippocampal sections were incubated with ACSF saturated with the above gas mixture at 32.0 ± 0.5 °C for 2–3 h. During the experiment, a single brain slice was immersed in ACSF saturated with the gas mixture (4 mL/min, 30 °C) and placed on an 8 × 8 array of the MED system (microelectrode size, 50 mm × 50 mm; interelectrode distance, 450 mm), and the voltage signal was obtained by the MED64 system. Through stimulation of perforating fibers, the slope changes of the field excited synaptic potential (fEPSP) were recorded from the hippocampal DG area, and 30% of the stimulus intensity that could induce the maximum slope of the fEPSP was selected as the basic stimulus intensity. In the case of a stable baseline, high-frequency stimulation (100 Hz, 200 pulses) was performed to induce long-term potential (LTP). An increase in fEPSP slope over 20% of the basal level was employed as a criterion for induction success and was maintained for at least 30 min. In the experiment, fEPSP was recorded and observed for at least 120 min.

### 2.13. Statistical Analysis

Data are mean ± SEM, and were assessed with SPSS 15.0 (SPSS, Chicago, IL, USA). Two-way analysis of variance (ANOVA) with post hoc Bonferroni’s test was carried out to assess the differences in escape latency measured during the place navigation trials. The remaining experimental data in each group were compared by one-way ANOVA with post hoc Bonferroni’s test. In all experiments, *p* < 0.05 indicated a statistically significant difference.

## 3. Results

### 3.1. Minocycline Alleviates Sevoflurane-Induced Learning and Memory Impairments

In order to examine the role of minocycline in sevoflurane-associated postoperative cognitive dysfunction, we established a sevoflurane-induced POCD model via inhalation of 3% sevoflurane for 2 h daily for three days. During place navigation trials of the Morris water maze experiment, escape latency was markedly elevated in the Sev group compared with the Con and Sev + Min groups (Figure 1A). On the 6th day (spatial probe test), the data showed increased escape latency, fewer crossings of the initial platform, and decreased target quadrant time ratio, and the swimming path was disordered in the Sev group, which showed significant differences in comparison with control animals. However, mice intraperitoneally administered minocycline before sevoflurane exposure had a certain improvement in the number of original platform crossings, escape latency, target quadrant time ratio, and swimming path, with marked differences in comparison with the Sev group (Figure 1B–E). A similar cognitive impairment was found in the fear conditioning test. No significant differences in freezing percentage in each group were found in the cued fear conditioning test; however, the freezing level was decreased in the Sev group compared with the control and Sev + Min groups in the cued fear memory test (Figure 1G,H). The above results suggested that multiple sevoflurane exposures impaired cognitive function in aged mice, and minocycline pretreatment suppressed learning and memory impairments associated with sevoflurane.

**Figure 1 brainsci-13-00512-f001:**
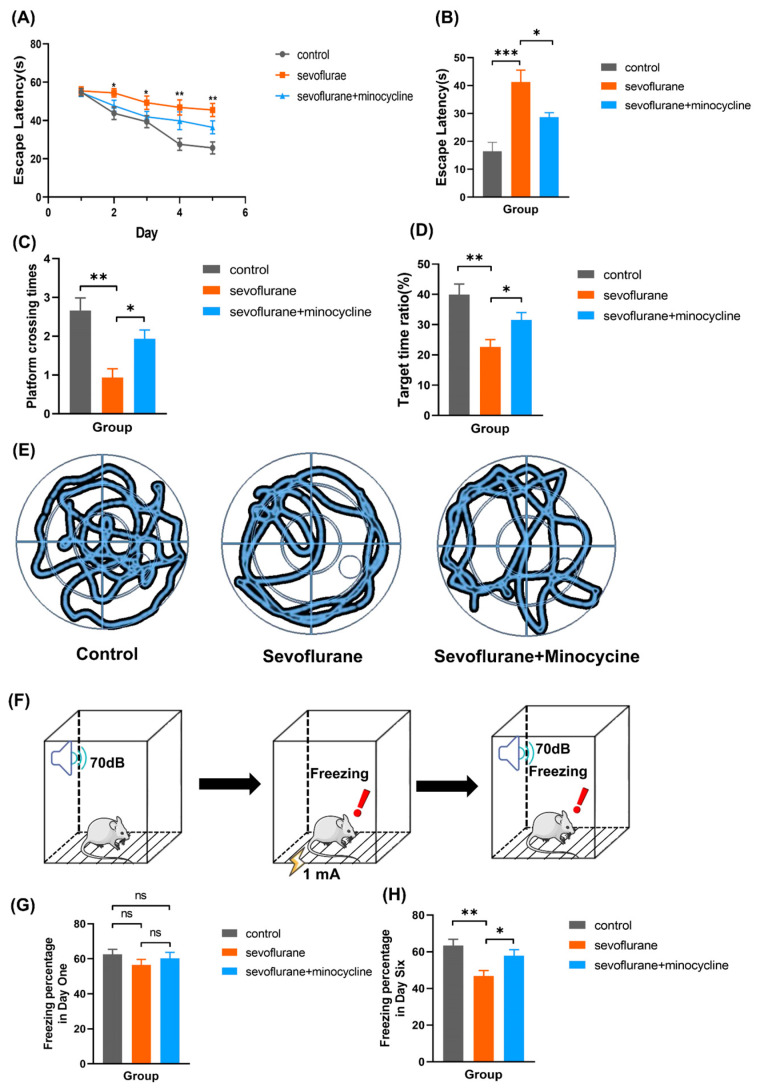
Minocycline alleviates learning and memory impairments associated with sevoflurane: (**A**) Escape latency (time required to find the platform) during the MWM training on days 1–5; (**B**) Escape latency in the space exploration experiment; (**C**) Platform crossing times during the space exploration test; (**D**) Target time ratio during the space exploration experiment; (**E**) Typical swimming track in each group; (**F**) Schematic diagram of the fear conditioning test; (**G**) Freezing percentage on day 1; (**H**) Freezing percentage on day 6. Data are mean ± SEM (*n* = 10). ns, no significance; * *p* < 0.05, ** *p* < 0.01, *** *p* < 0.001.

### 3.2. Minocycline Attenuates Sevoflurane-Induced Synaptic Plasticity Impairment

Synaptic plasticity is the basis of learning and memory; however, sevoflurane could impair synaptic plasticity by a variety of mechanisms [4,23]. To assess minocycline’s ameliorative effect on sevoflurane-associated synaptic plasticity impairment, synaptic plasticity was examined in acute brain slices by electrophysiological analysis (Figure 2A). The average fEPSP slope increased to 156.54 ± 2.86% (control group) and 157.22 ± 1.83% (sevoflurane + minocycline group) immediately after high-frequency stimulation (HFS) from baseline (100%) and was maintained, indicating long-term potentiation (LTP) was successfully induced in the hippocampus. Meanwhile, the fEPSP slope was much lower in the sevoflurane group (123.8 ± 2.67%, Figure 2B,C). To explore the possible reasons for the above changes, synapse-associated proteins were detected in hippocampal samples and HT-22 cells. We observed that postsynaptic density protein 95 (PSD95) and synapsin 1 (Syn-1) were significantly downregulated after sevoflurane exposure in both hippocampal extracts and HT-22 cells according to Western blotting data; nevertheless, minocycline pretreatment significantly enhanced PSD95 and Syn-1 protein expression in the hippocampal region and HT-22 cells (Figure 2D–G). These findings suggested minocycline attenuated sevoflurane-related impairment of synaptic plasticity.

### 3.3. Minocycline Alleviates Sevoflurane-Induced Neurogenesis Dysfunction

Neurogenesis is a process involving neural stem cells transforming into mature neurons and connecting with original neurons after proliferation, migration, and formation [24]. Thus, neurogenesis has a strong association with synaptic plasticity. Emerging evidence indicates sevoflurane has an inhibitory effect on hippocampal neurogenesis [25,26]. To confirm this notion and examine the therapeutic effect of minocycline, we used the nucleotide analog BrdU to label neural stem cells to observe the effects of sevoflurane on neurogenesis. As shown in Figure 3A,C, 24 h after sevoflurane exposure, BrdU-labeled cells were markedly decreased in the sevoflurane group in comparison with the control group, while cells pretreated with minocycline had remarkably increased BrdU-positive cells in comparison with the sevoflurane group. These findings suggested that 3% sevoflurane could lead to neural stem cell proliferative dysfunction and that minocycline alleviated this impairment. Twenty-one days after sevoflurane exposure, BrdU and NeuN co-staining was carried out to observe neural stem cell formation. In comparison with control cells, sevoflurane exposure starkly reduced the amounts of BrdU- and NeuN-positive cells in the hippocampal dentate gyrus; meanwhile, minocycline pretreatment markedly elevated the amounts of BrdU- and NeuN-labeled cells (Figure 3B,D). The above results suggested that minocycline alleviated sevoflurane-induced neurogenesis dysfunction.

### 3.4. Minocycline Suppresses Hippocampal Apoptosis Induced by Sevoflurane

Apoptosis is critical for multiple neurodegenerative processes [27]. Studies have suggested that sevoflurane can induce neuronal apoptosis [28,29]. To confirm this notion and examine the therapeutic effect of minocycline, we conducted TUNEL and immunoblot experiments. As shown in Figure 4A, in comparison with the control and minocycline groups, which were exposed to only 95% O_2_ and 5% CO_2_, the sevoflurane group had starkly elevated HT-22 cell apoptosis; however, upon minocycline pretreatment, the number of apoptotic HT-22 cells after sevoflurane exposure was reduced in comparison with the sevoflurane group, especially after treatment with high concentrations of minocycline (50 μM). Subsequently, apoptosis-associated proteins were detected. Similarly, in HT-22 cells, sevoflurane significantly upregulated proapoptotic proteins, including cleaved caspase 3, caspase 3, and Bax, and downregulated the antiapoptotic protein Bcl-2 compared with the control and minocycline groups. Treatment with moderate and high concentrations (10 μM and 50 μM) of minocycline before sevoflurane exposure significantly reversed the above changes in apoptosis-associated proteins (Figure 4B,D,F). Similar changes were also observed in hippocampal tissue specimens (Figure 4C,E,G). The above results suggested that sevoflurane exposure could induce hippocampal apoptosis and that minocycline suppressed hippocampal apoptosis induced by sevoflurane.

### 3.5. Minocycline Suppresses Sevoflurane-Associated Microglial Activation to the M1 Stage and Reduces Proinflammatory Cytokine Production

Increasing evidence suggests that minocycline is the most effective tetracycline derivative for neuroprotection [30]. Due to the neuroprotective effect of minocycline, we investigated whether it exerts a neuroprotective effect on sevoflurane-induced neuroinflammation. CD68 is a biomarker of the activation (proinflammatory) state in microglia. As shown in Figure 5A, sevoflurane significantly upregulated CD68 in BV2 cells, with significantly brighter red fluorescence, compared with the control and minocycline groups. Nevertheless, minocycline pretreatment before sevoflurane exposure significantly downregulated CD68 expression in BV2, and the red fluorescence was significantly attenuated, especially in the groups pretreated with the middle and high concentrations of minocycline. Similar results were observed in Western blot experiments (Figure 5B,C). Then, real-time quantitative PCR was carried out to examine the mRNA expression of proinflammatory cytokines. Unsurprisingly, sevoflurane exposure significantly increased the mRNA amounts of M1-type cytokines (iNOS and IL-1β) in comparison with the control and minocycline groups, while minocycline pretreatment before sevoflurane exposure markedly decreased iNOS and IL-1β mRNA amounts, particularly in the groups pretreated with the middle and high concentrations of minocycline (Figure 5D,E). The above results indicated that sevoflurane exposure could activate microglia, leading to neuroinflammation, while minocycline exerted a neuroprotective effect by suppressing sevoflurane-induced microglial activation to the M1 stage and reducing proinflammatory cytokine production.

### 3.6. Minocycline Alleviates Sevoflurane-Induced Neuroinflammation via Notch Signaling Suppression

According to the above data, we hypothesized that sevoflurane-induced neuroinflammation plays a dominant role in the observed changes. Thus, it is particularly important to unveil the underpinning molecular mechanisms behind sevoflurane-associated neuroinflammation. Recent studies have consistently suggested that the Notch signaling pathway has a strong association with inflammation [31,32]. To unveil the underlying molecular mechanisms of sevoflurane-induced neuroinflammation, immunofluorescence, and Western blot experiments were performed. As shown in Figure 6A, sevoflurane significantly upregulated Notch pathway-associated proteins in BV2 cells, including Notch1, cleaved Notch1, and Hes1, with significantly brighter red fluorescent signals compared with the control and minocycline groups. Nevertheless, minocycline pretreatment before sevoflurane exposure significantly downregulated Notch1, cleaved Notch1, and Hes1 in BV2 cells, and red fluorescent signals were significantly attenuated, especially in the groups pretreated with the middle and high concentrations of minocycline. Furthermore, Notch1, cleaved Notch1, and Hes1 protein amounts were dramatically decreased in hippocampal specimens and BV2 cells from the sevoflurane group according to Western blotting. Notably, minocycline treatment before sevoflurane exposure significantly downregulated Notch1, cleaved Notch1, and Hes1 proteins in the hippocampal region as well as BV2 cells (Figure 6B–G). These findings suggested that minocycline alleviated sevoflurane-associated neuroinflammation through inhibition of the Notch signaling pathway.

## 4. Discussion

In this work, we examined whether minocycline injection attenuates sevoflurane-associated neuroapoptosis and neuroinflammation in aged mice to alleviate cognitive dysfunction and determined the role of Notch signaling in minocycline suppression of sevoflurane-associated neuroinflammation. As expected, we found that exposure to 3% sevoflurane for 2 h a day for 3 consecutive days induced cognitive dysfunction in aged mice. Further investigation revealed that sevoflurane caused learning and memory impairments by inducing apoptosis, neuroinflammation, neurogenic dysfunction, and synaptic plasticity impairment. Minocycline significantly alleviated sevoflurane-induced cognitive dysfunction by exerting anti-inflammatory and antiapoptotic effects, which involved Notch signaling in aged mice.

As early as 1955, Professor Bedford first reported the clinical phenomenon that patients show such features as reduced interest in family, forgetting things, and indifference to the outside world after surgery and anesthesia. He defined them as “adverse brain effects of anesthesia in the elderly” [33]. As a common postoperative neurological complication, POCD brings many adverse effects to patients. According to a study performed in the United States in 2020, cognitive dysfunction occurring within one year after surgery adds approximately USD 17,275 to the patient’s health care expenditures [34]. In addition, POCD increases the incidence rates of other complications and prolongs hospitalization, but also carries a risk of developing into dementia [35]. Indeed, POCD has always had a clinically significant prevalence. A high incidence of cognitive impairment is observed in common surgical procedures in the elderly such as coronary artery bypass graft surgery and total hip arthroplasty. A study performed in the year 2019 showed that one week after coronary artery bypass graft surgery, 71% of patients experienced a decline in cognitive function and three months post-surgery, about 47% of cases still experienced changes in cognitive behavior [36]. A review showed that in individuals administered elective hip or knee arthroplasty, POCD had incidence rates of 19.3% and 10% at one and three months after surgery, respectively [37]. Anesthetic administration may represent a major risk factor for POCD. In 1997, a clinical trial by Galinkin et al. demonstrated that sevoflurane can decrease learning and memory function [38]. Since then, multiple reports have examined the association of sevoflurane with cognition. With the deepening of research, the mechanism of sevoflurane-induced cognitive impairment is constantly being revealed. Liang et al. found that sevoflurane induces learning and memory deficits in aged mice by reducing the plasma Aβ1-40 concentration and upregulating RAGE at both the transcriptional and translational levels in the brain [39]. Dysregulated apoptosis contributes to multiple human disorders, e.g., neurodegenerative disorders, and multiple studies have revealed that inappropriate apoptosis is strongly related to Alzheimer’s disease (AD) [40], Parkinson’s disease (PD) [41], and amyotrophic lateral sclerosis (ALS) [42]. Chen et al. found that neuronal apoptosis mediated by endoplasmic reticulum (ER) stress might be involved in memory impairment associated with sevoflurane in aging rats [43], which is consistent with our findings. As demonstrated above, sevoflurane triggered apoptosis, upregulating apoptosis-associated proteins (cleaved caspase 3, caspase 3, and Bax) and downregulating antiapoptotic protein Bcl-2. Shen and collaborators demonstrated that sevoflurane induces neuroinflammation in young mice but not in adult mice by increasing calcium amounts to upregulate TNF-α and IL-6 through the nuclear factor-κB pathway [44]. Similarly, Dong et al. showed that sevoflurane causes cognitive decline by increasing microglia-regulated neuroinflammatory reactions in a rat model by decreasing PPAR-γ activity in the hippocampus [45], indicating that neuroinflammation is an important mechanism of sevoflurane-induced cognitive impairment. Notch signaling is expressed in the majority of cells. With the help of ADAM metalloprotease and the γ-secretase complex, Notch receptor cleavage occurs, and the Notch intracellular domain (cleaved Notch1, an activated fragment) is released. Then, cleaved Notch1 is translocated to the nuclear compartment and binds to CSL (RBP-J) for transcriptional regulation of downstream genes such as members of the HES family [46,47]. Recent studies have consistently suggested that Notch signaling has a strong relationship with inflammation. Qian et al. found that Notch signaling pathway activation is critical for the differentiation of A1 astrocytes after spinal cord injury (SCI) [31]. Similarly, Wu et al. found that Notch signaling regulates the activation of microglia and inflammatory responses in rats with experimental temporal lobe epilepsy [32]. In the current study, we found that sevoflurane polarized microglia to a proinflammatory state (M1 state) by enhancing Notch signaling, producing inflammatory cytokines, and causing neuroinflammation. Moreover, we found that neuroinflammation impaired hippocampal neurogenesis, leading to reduced synaptic plasticity and cognitive impairment, corroborating previous studies [48,49]. Minocycline was shown to protect the nervous system through anti-inflammatory effects [50]. Due to its anti-inflammatory properties, multiple reports have focused on minocycline’s therapeutic effect on cognitive dysfunction. However, in 2023, Takazawa et al. examined minocycline’s effect on POCD in elderly individuals administered total knee arthroplasty and found that 200 mg of minocycline daily from the day before surgery until the seventh day after surgery did not decrease POCD incidence, contradicting many preclinical studies [51]. The negative results of this study may be related to the short dosing period. Indeed, a study in 2021 showed that the duration of postoperative neuroinflammation in aged mice may be as long as 14 days [52]. In 2015, Tian and collaborators reported for the first time that minocycline reduces sevoflurane-associated neuroapoptosis and inflammation by suppressing sevoflurane-associated Aβ buildup and NF-κB pathway activation in a hippocampal sample from aged rats [16]. Two years later, Tian et al. again found that minocycline might protect from sevoflurane-associated cell damage through Nrf2-dependent antioxidation and NF-κB pathway suppression [53]. Both studies conducted by Tian et al. revealed that minocycline attenuates sevoflurane-related cognitive dysfunction through anti-inflammatory effects, but how minocycline suppresses sevoflurane-induced neuroinflammation remains undefined. The present study found that minocycline attenuates sevoflurane-induced POCD in aged mice by suppressing hippocampal apoptosis, neuroinflammatory, neurogenesis dysfunction, and synaptic plasticity impairment. Moreover, we found that Notch signaling participated in sevoflurane-associated neuroinflammation and that minocycline protected against neuroinflammation by regulating the Notch signaling pathway.

The limitations of this study should be mentioned. First, we focused on minocycline’s protective effects on short-term sevoflurane-associated cognitive dysfunction. However, studies have shown that sevoflurane has the potential to cause long-term cognitive impairment as well [54,55], so it is still worth investigating whether minocycline improves long-term cognitive dysfunction induced by sevoflurane. Secondly, how sevoflurane activates Notch signaling also deserves further investigation.

## 5. Conclusions

This study indicates that sevoflurane induces cognitive dysfunction in aged mice by causing apoptosis, neuroinflammation, neurogenesis dysfunction, and impaired synaptic plasticity. Minocycline attenuates sevoflurane-induced synaptic plasticity impairment through antiapoptotic and anti-neuroinflammatory effects, so as to improve cognitive dysfunction, and its anti-inflammatory effect is at least partially achieved via the Notch signaling pathway. In future studies, we will continue to explore the mechanism of how sevoflurane activates Notch signaling and whether minocycline improves long-term cognitive dysfunction induced by sevoflurane. In summary, our study indicates that minocycline use is still expected to become a new treatment method to prevent POCD induced by sevoflurane in clinics.

## Figures and Tables

**Figure 2 brainsci-13-00512-f002:**
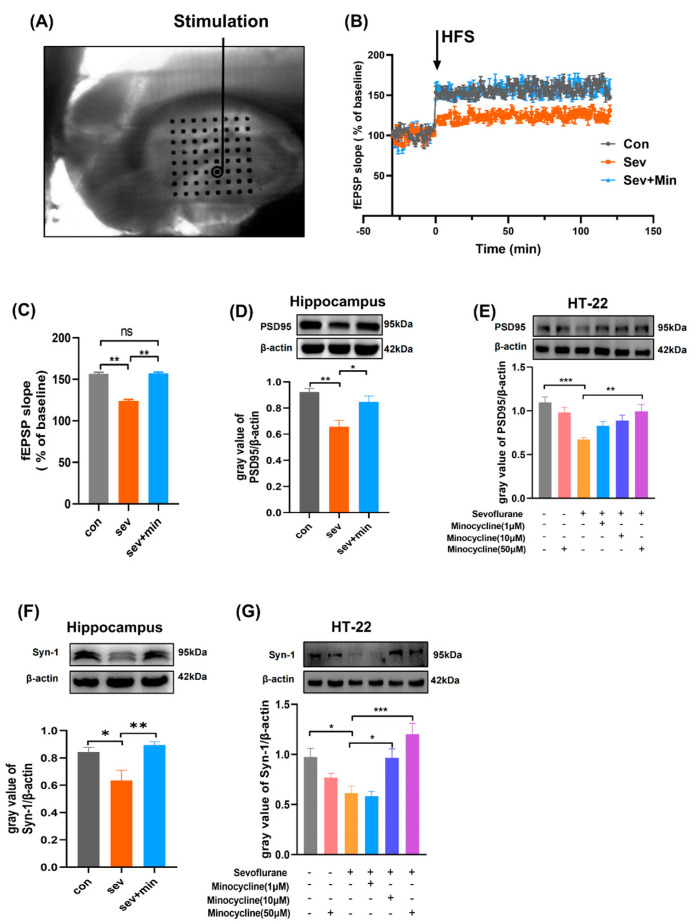
Minocycline attenuates sevoflurane-induced synaptic plasticity impairment. (**A**) Schematic representations of the stimulation sites in the hippocampal DG region and the microelectrode array (electrode size, 50 × 50 μm; interpolar distance of the electrode, 150 μm). HFS induced LTP in the DG region of the hippocampus. fEPSP slopes (**B**,**C**) in DG neurons in the hippocampus induced by the Schaffer collateral pathway. Data are mean ± SD (*n* = 3). ns, no significance; ** *p* < 0.01. (**D**–**G**) WB detection of PSD95 and Syn-1 expression levels in HT-22 cells and hippocampal tissue specimens. β-actin was utilized for normalization. ImageJ was employed for quantitation. Data are mean ± SEM. * *p* < 0.05, ** *p* < 0.01, *** *p* < 0.001.

**Figure 3 brainsci-13-00512-f003:**
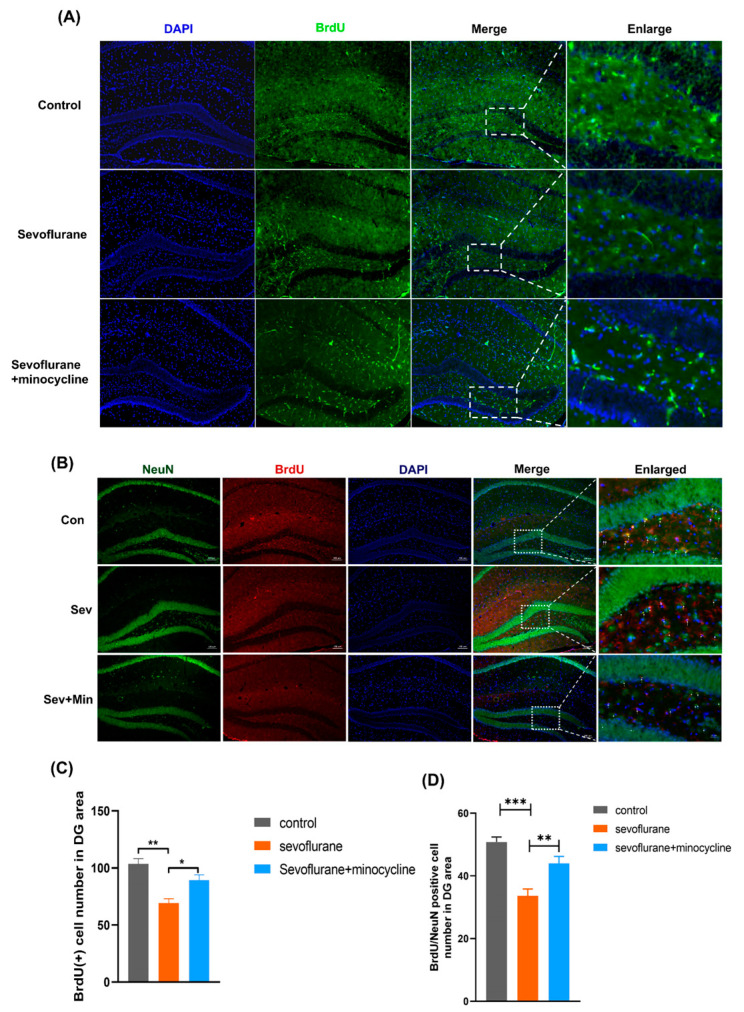
Minocycline alleviates sevoflurane-induced neurogenesis dysfunction: (**A**) Proliferation of BrdU-labeled hippocampal neural stem cells in various groups; (**B**) Maturation of BrdU-positive neural stem cells in the hippocampus of each group; (**C**) Numbers of BrdU-labeled cells in the DG region of the hippocampus. BrdU, green; nuclei, blue; (**D**) Numbers of BrdU and NeuN co-stained cells in the DG region of the hippocampus. BrdU, red; NeuN, green; nuclei, blue. Data are mean ± SEM. * *p* < 0.05, ** *p* < 0.01, *** *p* < 0.001.

**Figure 4 brainsci-13-00512-f004:**
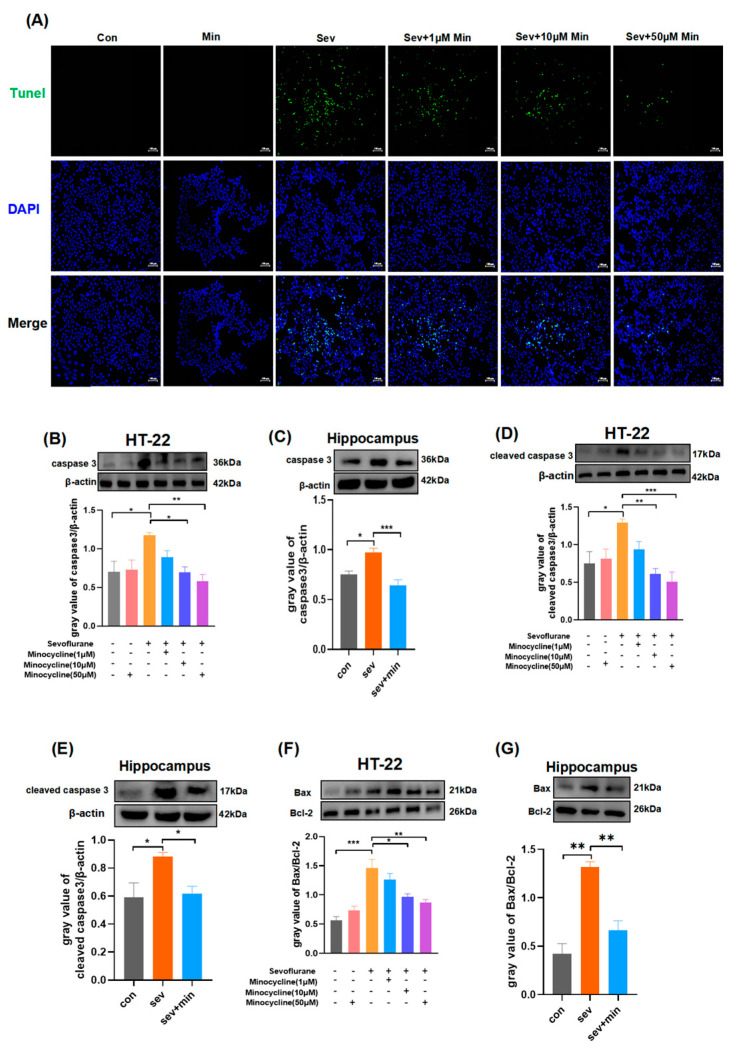
Minocycline suppresses the hippocampal apoptosis induced by sevoflurane: (**A**) TUNEL detection of neuronal apoptosis. TUNEL-positive, green; nuclei, blue. Scale bar, 100 μm; (**B**–**G**) Immunoblot detection of caspase-3, cleaved caspase-3, Bax, and Bcl-2 in HT-22 cells or hippocampal tissue samples. β-actin was employed for normalizing data quantitated by densitometry with ImageJ. Data are mean ± SEM. * *p* < 0.05, ** *p* < 0.01, *** *p* < 0.001.

**Figure 5 brainsci-13-00512-f005:**
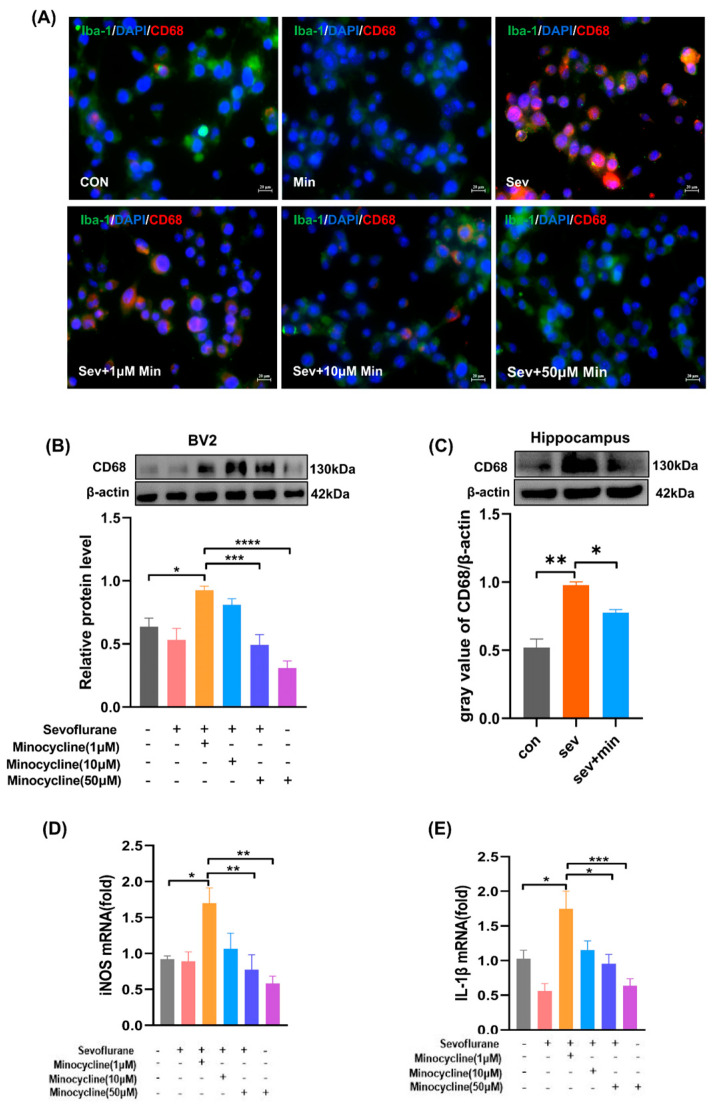
Minocycline suppresses sevoflurane-induced microglial activation to the M1 stage and reduces proinflammatory cytokine production: (**A**) Cell immunofluorescence to detect microglial activation. Iba-1, green; CD68, red; nuclei, blue. Scale bar, 20 μm; (**B**,**C**) Immunoblot detection of CD68 in BV2 cells and hippocampal tissue specimens. β-actin was employed for normalizing data quantitated by densitometry with ImageJ software. BV2 cells were collected for total RNA isolation, and (**D**) iNOS and (**E**) IL-1β gene expression levels were assessed. Data are mean ± SEM. * *p* < 0.05, ** *p* < 0.01, *** *p* < 0.001, **** *p* < 0.0001.

**Figure 6 brainsci-13-00512-f006:**
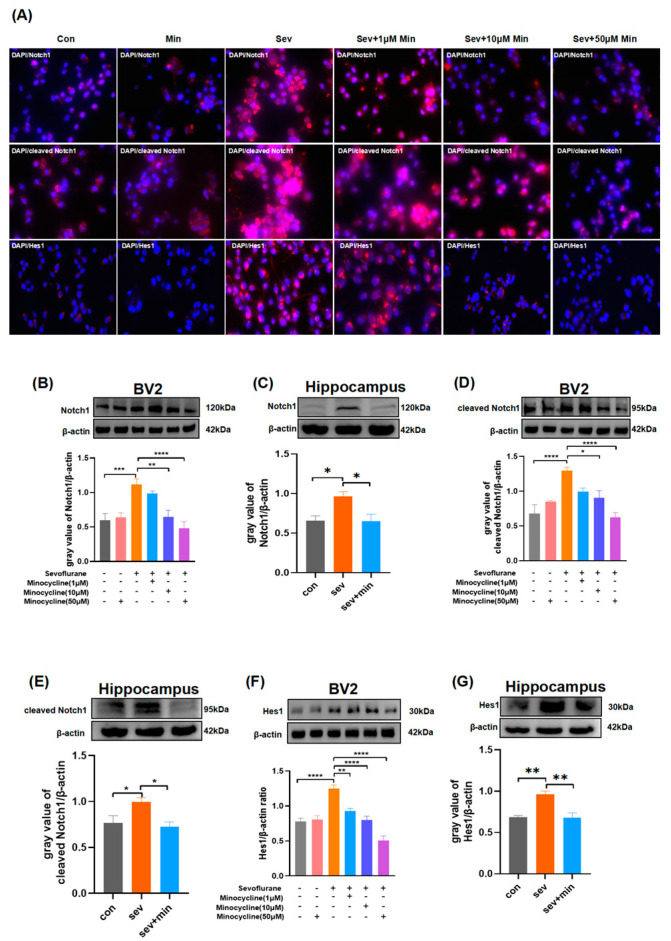
Minocycline alleviates sevoflurane-induced neuroinflammation via Notch signaling suppression: (**A**) Cell immunofluorescence was performed to assess Notch signaling-associated proteins in BV2 cells of each group. Scale bar, 20 μm; (**B**–**G**) Immunoblot detection of Notch1, cleaved Notch1, and Hes1 in BV2 cells and hippocampal tissue specimens. β-actin was utilized for normalizing data quantitated by densitometry with ImageJ. Data are mean ± SEM. * *p* < 0.05, ** *p* < 0.01, *** *p* < 0.001, **** *p* < 0.0001.

**Table 1 brainsci-13-00512-t001:** Primary antibodies utilized for immunofluorescence.

Antibody	Source	Catalog Number	Dilution	Specificity
Iba-1	Thermofisher scientific	PA5-27436	1:150~1:200	Mouse
CD68	CST	97778S	1:150~1:200	Rabbit
BrdU	Abcam	Ab8955	1:150~1:200	Mouse
NeuN	Abcam	Ab177487	1:150~1:200	Rabbit
Notch1	Proteintech	20687-1-AP	1:150~1:200	Rabbit
Cleaved Notch1	Affinity Bioscience	AF5307	1:150~1:200	Rabbit
Hes1	Affinity Bioscience	DF7569	1:150~1:200	Rabbit

**Table 2 brainsci-13-00512-t002:** Primary antibodies used in Western blotting.

Antibody	Source	Catalog Number	Dilution	Specificity
Caspase3	Affinity Biosciences	AF6311	1:500	Rabbit
Cleaved caspase3	Affinity Biosciences	AF7022	1:500	Rabbit
Bcl-2	Affinity Biosciences	AF6139	1:500	Rabbit
Bax	Affinity Biosciences	AF0120	1:500	Rabbit
PSD95	Proteintech	20665-1-AP	1:500	Rabbit
Syn-1	Proteintech	20258-1-AP	1:500	Rabbit
Notch1	Proteintech	20687-1-AP	1:500	Rabbit
Cleaved Notch1	Affinity Biosciences	AF5307	1:500	Rabbit
Hes1	Affinity Biosciences	DF7569	1:500	Rabbit
β-actin	Affinity Biosciences	AF7018	1:1000	Rabbit

**Table 3 brainsci-13-00512-t003:** Primer sequences.

Primer	Forward (5′-3′)	Reverse (5′-3′)
iNOS	CCCTTCAATGGTTTACATGG	ACATGATCTCCGTGACAGCC
IL-1β	ACTCATTGTGGCTGTGGAGA	TTGTTCATCTCGGAGCCTGT
GAPDH	AACGACCCCTTCATTGACCT	TGGAAGATGGTGATGGGCTT

## Data Availability

All data supporting the findings of this study are available to the corresponding author upon request.

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
