# Peer review of "Minocycline Attenuates Sevoflurane-Induced Postoperative Cognitive Dysfunction in Aged Mice by Suppressing Hippocampal Apoptosis and the Notch Signaling Pathway-Mediated Neuroinflammation"

_brainsci, 2023, doi:10.3390/brainsci13030512_

Round 1
Reviewer 1 Report
Dear Sirs,
In the manuscript entitled “Minocycline attenuates sevoflurane-induced postoperative cognitive dysfunction in aged mice by suppressing hippocampal apoptosis and the Notch signaling pathway-mediated neuroinflammation, authors explored the role of the Notch signaling pathway in sevoflurane-induced neuroinflammation and the therapeutic effect of minocycline.
The work is interesting and the authors present a very good number of experiments and work, including with mice and cell cultures.
However, results from a clinical trial, published recently, indicate that minocycline is likely to have no preventive effect on POCD (2023, Prevention of Postoperative Cognitive Dysfunction by Minocycline in Elderly Patients after Total Knee Arthroplasty: A Randomized, Double-blind, Placebo-controlled Clinical Trial. Anesthesiology, 138(2), 172–183. https://doi.org/10.1097/ALN.0000000000004439).
The above publication may have great significance and take importance to the work submitted.
Nevertheless, this work can contribute to the understanding of the molecular mechanisms underlying the anti-inflammatory and antiapoptotic activities of minocycline in the central nervous system.
Comments:
1- How was the sevoflurane scheme of administration determined (2 hours a day, for 3 consecutive days)? How does it relate to POCD in humans?
2- For greater accuracy in comparing animals of different groups, control, and sevoflurane groups must be intraperitoneally injected with a vehicle (i.e. saline).
Can authors anticipate any influence on results due to this lack of appropriate control?
3- Briefly comment on the cell lines used (origin, used as a model of, etc.) to justify their usage.
4- Abbreviations ANP32A, C/EBPβ, Nuak1 should be explained
5- Material and methods: section 2.7- why authors considered needed to block endogenous peroxidase for immunofluorescent analysis?
6- It would be valuable to include references supporting the importance of POCD relative to work developed in other areas of the globe. The bibliography seems to be very much dependent on one geographical area.
Minor comments:
1- Abstract – “… (50 mg/kg, 24 i.p, one hour before sevoflurane exposed)…”
Should be exposure.
2- Figure 1 seems misplaced, appearing in the middle of Material and Methods. Although referred to in this section, this figure mainly provides results obtained in the behavioral tests.
3- On page 8: “On the 6th day (the spatial probe test), the data showed an increased escape latency, fewer crossing numbers of the original platform and a decreased target quadrant time ratio in the Sev group, which was significantly different compared with the control group, and the swimming path was disordered”
This sentence is confusing. Which animals presented disordered swimming?
Author Response
Dear Review:
Thank you for your time to review our manuscript entitled “Minocycline attenuates sevoflurane-induced postoperative cognitive dysfunction in aged mice by suppressing hippocampal apoptosis and the Notch signaling pathway-mediated neuroinflammation” (Manuscript ID: brainsci-2269257). Those comments are all valuable and very helpful for revising and improving our paper, as well as the important guiding significance to our researches. We have studied comments carefully and have made corrections which we hope meet with approval. According to the requirements of editor, we have revised the paper using the “Track Changes” function. The main corrections in the paper and the responses to the Reviewer’s comments are as follows:
1.In the manuscript entitled “Minocycline attenuates sevoflurane-induced postoperative cognitive dysfunction in aged mice by suppressing hippocampal apoptosis and the Notch signaling pathway-mediated neuroinflammation, authors explored the role of the Notch signaling pathway in sevoflurane-induced neuroinflammation and the therapeutic effect of minocycline. The work is interesting and the authors present a very good number of experiments and work, including with mice and cell cultures. However, results from a clinical trial, published recently, indicate that minocycline is likely to have no preventive effect on POCD (2023, Prevention of Postoperative Cognitive Dysfunction by Minocycline in Elderly Patients after Total Knee Arthroplasty: A Randomized, Double-blind, Placebo-controlled Clinical Trial. Anesthesiology, 138(2), 172–183. https://doi.org/10.1097/ALN.0000000000004439). The above publication may have great significance and take importance to the work submitted. Nevertheless, this work can contribute to the understanding of the molecular mechanisms underlying the anti-inflammatory and antiapoptotic activities of minocycline in the central nervous system.
Response 1: Please see the attachment.
2.How was the sevoflurane scheme of administration determined (2 hours a day, for 3 consecutive days)? How does it relate to POCD in humans?
Response 2: Please see the attachment.
3.For greater accuracy in comparing animals of different groups, control, and sevoflurane groups must be intraperitoneally injected with a vehicle (i.e. saline). Can authors anticipate any influence on results due to this lack of appropriate control?
Response 3: Please see the attachment.
4.Briefly comment on the cell lines used (origin, used as a model of, etc.) to justify their usage.
Response 4: Please see the attachment.
5.Abbreviations ANP32A, C/EBPβ, Nuak1 should be explained.
Response 5: Please see the attachment.
6.Material and methods: section 2.7- why authors considered needed to block endogenous peroxidase for immunofluorescent analysis?
Response 6: Please see the attachment.
7.It would be valuable to include references supporting the importance of POCD relative to work developed in other areas of the globe. The bibliography seems to be very much dependent on one geographical area.
Response 7: Please see the attachment.
8.Abstract – “… (50 mg/kg, 24 i.p, one hour before sevoflurane exposed)…” Should be exposure.
Response 8: Please see the attachment.
9.Figure 1 seems misplaced, appearing in the middle of Material and Methods. Although referred to in this section, this figure mainly provides results obtained in the behavioral tests.
Response 9: Please see the attachment.
10.On page 8: “On the 6th day (the spatial probe test), the data showed an increased escape latency, fewer crossing numbers of the original platform and a decreased target quadrant time ratio in the Sev group, which was significantly different compared with the control group, and the swimming path was disordered”. This sentence is confusing. Which animals presented disordered swimming?
Response 10: Please see the attachment.

Reviewer 2 Report
This is a well-structured article. The main question addressed by this research is the fact that Minocycline attenuates sevoflurane-induced postoperative cognitive dysfunction in aged mice by suppressing hippocampal apoptosis and the Notch signaling pathway mediated neuroinflammation.
The introduction gives the background of this study as it briefly describes the effect of surgery and anesthesia on the aged brains and the various pharmacological actions of Minocycline besides its main antibacterial action.
“Materials and Methods” section is descriptive enough. It refers to the animals used in the study, the procedures that were followed, the tests that were implemented and the statistical analysis that was performed during this study.
The results are quite interesting and, to my opinion, well presented with a lot of figures.
The discussion is well written, summarizing and discussing the main findings of the study and trying to correlate it with recent relative studies. I think that a paragraph summarizing the main limitations of the study will add to the scientific value of this paper.
Furthermore, the conclusions could be written in a more detailed manner, perhaps proposing some specific targets for future studies.
References, although relatively few, are relative to the subject.
English language and style are generally fine there are some minor issues that need to be addressed.
Author Response
Dear Review:
Thank you for your time to review our manuscript entitled “Minocycline attenuates sevoflurane-induced postoperative cognitive dysfunction in aged mice by suppressing hippocampal apoptosis and the Notch signaling pathway-mediated neuroinflammation” (Manuscript ID: brainsci-2269257). Those comments are all valuable and very helpful for revising and improving our paper, as well as the important guiding significance to our research. We have studied comments carefully and have made corrections which we hope meet with approval. According to the requirements of editor, we have revised the paper using the “Track Changes” function. The main corrections in the paper and the responses to the Reviewer’s comments are as follows:
1.This is a well-structured article. The main question addressed by this research is the fact that Minocycline attenuates sevoflurane-induced postoperative cognitive dysfunction in aged mice by suppressing hippocampal apoptosis and the Notch signaling pathway mediated neuroinflammation. The introduction gives the background of this study as it briefly describes the effect of surgery and anesthesia on the aged brains and the various pharmacological actions of Minocycline besides its main antibacterial action. “Materials and Methods” section is descriptive enough. It refers to the animals used in the study, the procedures that were followed, the tests that were implemented and the statistical analysis that was performed during this study. The results are quite interesting and, to my opinion, well presented with a lot of figures.
Response 1: Please see the attachment.
2.The discussion is well written, summarizing and discussing the main findings of the study and trying to correlate it with recent relative studies. I think that a paragraph summarizing the main limitations of the study will add to the scientific value of this paper.
Response 2: Please see the attachment.
3.Furthermore, the conclusions could be written in a more detailed manner, perhaps proposing some specific targets for future studies.
Response 3: Please see the attachment.
4.References, although relatively few, are relative to the subject.
Response 4: Please see the attachment.
5.English language and style are generally fine there are some minor issues that need to be addressed.
Response 5: Please see the attachment.

Reviewer 3 Report
The MS "Minocycline attenuates sevoflurane-induced postoperative cognitive dysfunction in aged mice by suppressing hippocampal apoptosis and the Notch signaling pathway-mediated neuroinflammation" is clear and well written. Authors provide a high quality and really interesting study.
I have just minor comments that authors should address.
Introduction:
Could you please clearly state the aims and hypothesis of this study
Methods:
could you please add the rational of using these specific type of cell lines
Results
fig 3: How did you count the cells? did you use a software? Please add the method in the corresponding section
Fig 4 the representative image of westernblot on BAX is not really clear, could you please provide a clearer one
In the figures presenting fluorescent microscopy images, I would suggest to enlarge the images, colors are fade and it's hard to distinguish the different markers
Discussion:
Please add the limitations of your study as well as the perspectives
Author Response
Dear Review:
Thank you for your time to review our manuscript entitled “Minocycline attenuates sevoflurane-induced postoperative cognitive dysfunction in aged mice by suppressing hippocampal apoptosis and the Notch signaling pathway-mediated neuroinflammation” (Manuscript ID: brainsci-2269257). Those comments are all valuable and very helpful for revising and improving our paper, as well as the important guiding significance to our research. We have studied comments carefully and have made corrections which we hope meet with approval. According to the requirements of editor, we have revised the paper using the “Track Changes” function. The main corrections in the paper and the responses to the Reviewer’s comments are as follows.
1.The MS "Minocycline attenuates sevoflurane-induced postoperative cognitive dysfunction in aged mice by suppressing hippocampal apoptosis and the Notch signaling pathway-mediated neuroinflammation" is clear and well written. Authors provide a high quality and really interesting study. I have just minor comments that authors should address. Introduction: Could you please clearly state the aims and hypothesis of this study.
Response 1: Please see the attachment.
2.Methods: could you please add the rational of using these specific type of cell lines.
Response 2: Please see the attachment.
3.Results: fig 3: How did you count the cells? did you use a software? Please add the method in the corresponding section.
Response 3: Please see the attachment.
4.Fig 4 the representative image of western blot on BAX is not really clear, could you please provide a clearer one.
Response 4: Please see the attachment.
5.In the figures presenting fluorescent microscopy images, I would suggest to enlarge the images, colors are fade and it's hard to distinguish the different markers.
Response 5: Please see the attachment.
6.Discussion: Please add the limitations of your study as well as the perspectives.
Response 6: Please see the attachment.

Round 2
Reviewer 1 Report
Dear Sirs,
I have now analyzed the authors' answers to the questions raised and concluded that they were all adequately responded to. Furthermore, the necessary and appropriate alterations were performed in the manuscript.
As such, I endorse the publication of this work in the Brain Sciences Journal.